# Fluid Mixing Nonequilibrium Processes in Industrial Piping Flows

## Mikhail Sukharev

Department of Applied Mathematics and Computer Modeling, National University of Oil and Gas «Gubkin University», 119991 Moscow, Russia; mgsukharev@mail.ru

**Abstract:** The flow of a multicomponent fluid through a pipeline system of arbitrary configuration is considered. The problem consists in determining the component composition of the fluid for each pipeline of the system based on the values of the concentration of the components throughout the entire set of measuring points, provided that there are no phase transitions. To solve the problem, mathematical models have been developed that, in principle, are suitable for pipeline systems of various functional purposes, the presentation is concretized and carried out in relation to gas transmission systems. The models are stochastic in nature due to measurement errors, which are considered random variables. The solution of the problem is reduced to the optimization of a quadratic function with constraints in the form of equalities and inequalities. The considered mixing processes do not depend on the regime parameters of the fluid flow. The processes are irreversible and non-equilibrium. A criterion is introduced that characterizes the degree of closeness of a multicomponent mixture to an equilibrium state. The criterion is analogous to entropy in thermodynamic processes. A numerical example of calculating the distribution of a three-component mixture is given. The example illustrates the feasibility of the proposed computational procedures and gives an idea of the distribution of the component composition and the change in «entropy» along the directions of pumping of the gas supply system.

**Keywords:** multi-component flows; gas transmission systems; non-equilibrium processes; mathematical models; maximum likelihood method; calorific value; entropy

## 1. Introduction

Gas enters the Unified Gas Supply System (UGSS) of the Russian Federation from different sources and differs in its composition, although methane remains the main component. The share from Russian fields ranges from 90% to 98%. By way of example, let us consider composition analysis of gas transported via a gas line in the North of Siberia (gas composition in molar fractions, %: methane 93.2; ethane 4.1; propane 1.30; i-butane 0.19; n-butane 0.18; neo-pentane 0.00l3; i-pentane 0.030; n-pentane 0.02l; hexane 0.0084; carbon dioxide 0.32; nitrogen 0.59; oxygen—less than 0.005; helium—0.0096; hydrogen 0.0018; water vapour—0).

In addition to gas and gas condensate fields, UGSS supply sources include oil fields (associated gas), gas processing plants, underground storage facilities, and import supplies. From plants, oil fields, underground storages in depleted oil fields there arrives rich gas with a greater share of heavy hydrocarbons and, therefore, higher calorific value. In the process of transportation, the concentrations change with the mixing of fluids at the junction points of the pipelines. Composition and calorific value of gas transported via the UGSS are measured by samples taken at gas metering stations (GMS). These indicators can be measured periodically using gas cylinders or continuously if the necessary equipment is available at the measuring installation [1–3]. Along the length of each pipeline, the concentrations/calorific values do not change, so it is natural to associate them with a given pipeline

regardless of where the measuring point is. The problem is to estimate the concentration values for each pipeline based on the set of measurements. Any measurements are fraught with random errors and, therefore, the measured concentrations must be random values, so the concentration estimation models must be stochastic.

In the process of operational control, such parameters of the mode as pressure, flow rate, temperature are measured, and their changes and distribution in the directions of pumping are monitored. However, the calorific value turns out to be a not less important parameter, and tracking its distribution throughout the system should also be included in the number of dispatching tasks. Furthermore, in pipeline systems, emergency situations occur, accompanied by the injection of substances that reduce the quality of the transported product. Tracking the spread of the "pollutant" is also one of the problems of monitoring the component composition.

The paper considers a large-scale gas transmission system (GTS) and sets the task of finding for it the most reasonable ways to estimate the component composition (CC) for each direction of pumping. The component composition uniquely determines the calorific value of the gas. Caloric content can be calculated either knowing the CC, or setting this problem as the main one and solving it using the same methods as the CC calculation.

Here is an incomplete list of problems for the solution of which it is necessary to know the distribution of the CC of the fluid through the directions of pumping.

(a)　Settlements between suppliers and consumers are based on energy measurements and with regard to supplied gas calorific value. Differences in consumer requirements to gas calorific value make suppliers interested in calorific value management, which can be achieved by optimizing gas system flows.

(b)　Necessity of the UGSS metrological support enhancement. With the gradual depletion of reserves in the Nadym-Purtaz region, deeper deposits will be developed, which will increase methane homologues quantity in the total amount of hydrocarbons produced. The problem of rational distribution of rich gas will become more important. It will be necessary to increase the accuracy of determining CC/caloric value of gas supplied to consumers. The economic significance of the models considered in the paper will grow. Such models will be required to compare UGSS metrological equipment development and modernization options.

(c)　The problem of placement of gas chemical complexes. Hydrocarbons of the methane homologous series are a more valuable raw material for gas chemistry than methane. It is economically feasible to separate them from natural gas for use in gas chemistry. The choice of gas chemical facilities location presents a serious problem and amounts to two options: whether to place the complexes in proximity to gas fields or to industrial centers. Each of the options has its pros and cons. An analysis of the gas component distribution within the UGSS is indispensable for making the right decision. To make the right decision, it is necessary to analyze the component composition of gas through the UGSS pipelines.

(d)　Calculation of dew point temperature for water and hydrocarbons. Failure to comply with regulatory requirements for their parameters is fraught with an increase in the risk of hydrate and condensate formation, and for export deliveries it entails penalties. In our opinion, the ideas of evaluating the component composition will also be suitable for solving these problems, but under complicating circumstances: the need to consider the phase space with the inclusion of pressure and temperature.

(e)　Provide some modification to account for technological specifics, and herein the proposed methods are also applicable to other pipeline systems: oil trunk transportation (compounding oils from different fields, monitoring the content of hydrogen sulfide), water supply (compliance with water quality standards), etc.

Thus, studying the CC distributions of natural gases is very important in practice. The significance of the problem was also noted by Tevyashev et al. [4]. A formalized model and methods for its solution

for two-component mixtures were first proposed by Sukharev, Kislenko et al. [2,3]. As shown above, the tasks of determining the CC arise both in operational control and in planning the development of the UGSS. Papers [5,6] should be considered the first steps in the study of the problems. In this article, the following steps have been taken in the same direction. First, not two-component, but three-component mixtures are considered. This led not only to an increase in the number of sought variables. The non-equilibrium and irreversibility of mixing processes lead to the introduction into the model of a large number of constraints in the form of inequalities. Some of them turn out to be "needless", that is, they are automatically executed even in solutions when they are not entered into the model. In problems for two-component mixtures, the significance of non-equilibrium constraints was not manifested. In the flows of three-component mixtures (a) the number of unknowns increases (compared to two-component ones), (b) one has to deal with situations where unknowns differ by orders of magnitude, (c) among the non-equilibrium constraints, significant constraints appear as a rule.

To overcome the obstacles that arise, we propose various methods. They are tested in computational experiments. In particular: (a) the nested solutions method—a subject-oriented iterative procedure that allows one to gradually introduce significant non-equilibrium constraints into the model, thereby reducing the dimension of mathematical programming problems at the stages of an iterative procedure; (b) a new approach to solving problems by non-standard introduction of unknowns (differing by orders of magnitude), eliminating (or at least smoothing out) the problem of the appearance of unknowns differing by orders; (c) instead of concentrations, which often turn out to be small quantities of different orders, the unknowns are the jet intensities—quantities that usually differ from each other by no more than an order of magnitude; (d) a criterion for the information content of metrological support, which makes it possible to judge the possibilities of assessing the CC and/or the caloric content of the fluid, depending on the number of measurement points and their location in the pipeline system; (e) quantitative characteristic of the degree of non-equilibrium of the mixing process (this is an analogue of entropy in classical thermodynamics, however, unlike the latter, it is not defined on a continuum, but on a discrete set–cuts of the pipeline system graph).

When determining the component composition/calorific value, it is assumed that the total flow rate of the fluid for each direction of pumping is known, that is, the flow distribution of the fluid—a mixture of gases–is calculated in advance and serves as initial information for determining the flow rates of each component. The inclusion of the flow distribution of the fluid in the initial information essentially means the decomposition of the problem. At the highest level of the hierarchy is the calculation of fluid flow rates by pumping directions. This calculation is carried out by standard methods with a detailed account of technology (structure, operating parameters: pressure and temperature, technical condition of equipment, etc.). An exact knowledge of the distribution of the component composition is not required in this case. The study of the component composition is the next level of the hierarchy, the subject of this work. Decomposition is also possible because the processes associated with phase transitions are not considered. Phase transitions during the operation of gas supply systems are manifested in the form of condensate–water or hydrocarbons–and hydrate formation. In our opinion, the construction of models for the distribution of CC/calorific value taking into account phase transitions is a matter for the near future. However, decomposition as implemented below will not be valid. When assessing the distribution of the CC, it is necessary to take into account the parameters of the gas flow, at least the temperature, as well as the models of condensate precipitation [7–9]. It is very likely that the results obtained in the field of thermodynamics of mixtures [10] can be useful here. Models of distribution of CC/calorific value, along with the developed apparatus of multimodal techniques [11,12], should, in our opinion, be used in the development of computer technologies for monitoring and control of large pipeline systems at the highest level of the hierarchy.

The rest of the manuscript is organized as follows. Section 2 describes the research methods used in the work. In particular, the specific features of irreversible and non-equilibrium processes of gas components mixing during their transportation are discussed (Section 2.1). The mathematical

formulization of the problem is presented in two versions: in the first, the unknown quantities are the flow rates of the components (Section 2.2); in the second, the intensity of the "jets" (Section 2.5). The conditions for the non-equilibrium of the mixing processes are derived (Section 2.3). The computational aspects of the procedures for estimating the CC of the fluid are discussed (Sections 2.4 and 2.5). Section 3 (results) provides an example of calculating a system whose graph contains 11 nodes and 13 arcs (transporting directions). An indicator of non-equilibrium of mixing processes is introduced and illustrated by an example. Section 4 contains a discussion of the results.

## 2. Methods

### 2.1. Non-Equilibrium Mixing Processes (Technological Aspect)

Successful solution of this task depends largely on how well the computational model is constructed. When calculating large gas supply systems, they often resort to network reduction and replace several parallel gas pipelines with a single arc of the calculation scheme. For this reason, when describing the model, the term transportation direction is more justified, than a gas pipeline, since an element of the calculation scheme can correspond to a pipeline system, and not to a single pipeline. When solving the task of gas mixture flow distribution through the system, the network reduction procedure should be carried out with caution. Questions arise when fluids with different concentrations are mixed at a junction point of the calculation scheme. Can we assume that the concentrations of components in all pipelines diverging from this junction point are equal? Concentration equality would take place if mixing processes were equilibrium ones. In fact, generally speaking, they do not possess the quality of equilibrium. The gas flow rates through pipelines of GTS are quite large and, as observations on large-scale systems show, complete mixing is observed only with a small load in the pipelines. The concentration values along the output lines depend on the local configuration of pipelines at the junctions. It is not possible to take into account such technological details in the aggregated scheme, since the geometric dimensions of the pipeline junction are incomparably smaller than the lengths of the pipelines. (Note that for this reason, attempts to build models of the CC distribution, taking into account the regime parameters of the flow, primarily pressure, are doomed to failure. The only acceptable way is the way we have chosen to take into account all simultaneous measurements of CC). So, to calculate the concentrations, the design scheme should be disaggregated so that the model makes it possible to reflect the difference in concentrations in the arcs emerging from the connection node. Suppose, for example, that two pipelines or two systems of parallel pipelines (pipeline corridors) converge at the junction, and the output is a multi-pipe corridor. Is it possible to represent this corridor as one arc in the aggregated scheme? If gases of different composition enter through the input lines, and it is known from observations of real regimes that the concentrations on the output lines differ, then this is impossible.

The mixing of components occurs for two reasons: due to diffusion and flow turbulence. The latter reason is decisive at the flow rates characteristic of GTS operation. Observations of the GTS regimes show that the processes of mixing natural gas components are isothermal. These processes are irreversible and non-equilibrium. Complete mixing is usually not achieved, which should be considered an experimentally established fact.

Irreversible non-equilibrium processes are studied by thermodynamics, physical chemistry, kinetic theory of gases. Irreversible non-equilibrium processes often take place during the production, transportation and processing of natural gas [13,14]. The range of these processes is expanding due to the emergence of new technologies. In this regard, we indicate several recent publications in the field of pipeline gas transportation. In [15] the problems of transporting a mixture of natural gas with hydrogen are considered, and in [16], those of transporting natural gas mixed with nitrogen.

Gas flow mixing schemes are traditionally considered in theoretical and technical thermodynamics [17–19]. However, usually the main attention is paid to the study of pressure and temperature of mixing flows. Evaluation of working capacity additional loss caused by irreversible

heat transfer between the mixing gases or by failure to use difference in flow pressures turns out to be especially important.

The kinetic theory of gases and, in many respects, physical chemistry work not with continual models, but with discrete models operating with the movement of molecules and the atomic structure of matter. They use the apparatus of quantum chemistry, statistical mechanics, and analytical dynamics [20].

In thermodynamics and in scientific disciplines that have arisen on its basis, a powerful apparatus has been developed for describing irreversible and non-equilibrium processes. The physical processes of fluid mixing in industrial pipeline systems considered in this paper are also irreversible and non-equilibrium, but their study does not require the use of this apparatus. First of all, this because the mixing processes are not accompanied by a change in the parameters of the regime: temperature and pressure—and phase transitions. But, nevertheless, to characterize the mixing processes we have proposed an indicator—an analogue of thermodynamic entropy (see Section 4).

*2.2. Task Formalization*

We will represent the structure of the pipeline system in the form of directed graph $G = (V, E)$, where $V$ is a set of nodes, and $E$ is a set of arcs. The arc in the aggregated scheme corresponds to transportation direction. By $m$ we denote the number of nodes, and by $n$ the number of arcs. (We use the terminology of the classical monographs of Christofides and Berge on graph theory [21,22]). The nodes are divided into 3 groups: inflows $V_{in}$, outflows $V_{out}$, junctions $V_{joint}$ (in Figure 1 $V_{in} = \{1, 2, 3, 4\}$, $V_{out} = \{9, 11\}$, $V_{joint} = \{5, 6, 7, 8, 10\}$). We assume that the inflows and outflows are connected to the graph by a single arc, outgoing for inflows (in Figure 1 $1 \equiv (1, 2)$, $2 \equiv (2, 6)$, $3 \equiv (3, 5)$, $4 \equiv (4, 5)$) and incoming for outflows (in Figure 1 $11 \equiv (8, 9)$, $13 \equiv (10, 11)$). A vector $\xi$ is set on the graph $G$. It represents mass flow rates of fluid consisting of several miscible components (in Figure 1 $\xi = \|170, 90, \ldots, 65, 135\|^{\mathsf{T}}$. This means that the flow distribution is calculated, as is customary in the practice of GTS operation, and the results of the calculation serve as the initial data for determining the CC for each direction of pumping. Naturally, the calculation is carried out taking into account the technical state of the system and the relationship between the regime parameters (pressure and temperature) of the gas flow. Since flow $\xi$ is initially known, all arcs $(i, j) \in E$ can be oriented along the flow, which means $\xi_{ij} \geq 0$. To simplify the notation, we assume that the number of components is 3; all formulas can obviously be rewritten for the case of an arbitrary number of components. $M = I, II, III$ is component number, $\xi_{ij}$ is flow rate of fluid, and $\eta_{ij}^M$ is component $M$ flow rate along the arc $(i, j)$.

By definition, $\xi_{ij} = \sum_{M=I}^{III} \eta_{ij}$. Values $\xi_{ij}$ satisfy material balance equations in all joint nodes $x_k$

$$\sum_{x_i \in \Gamma^{-1}(x_k)} \xi_{ik} - \sum_{x_j \in \Gamma(x_k)} \xi_{kj} = 0, \; x_k \in V_{joint}. \tag{1}$$

Hereinafter, $\Gamma(x_k)$ is a set of nodes into which arcs come from $x_k$, $\Gamma^{-1}(x_k)$ is a set of nodes from which arcs come into $x_k$ (Figure 2). Flow rates of each component of the mixture is also a vector $\eta^M = \|\eta_{ij}^M\|$, $(i, j) \in E$, $M = I, II, III$, components of that vector must satisfy of material balance conditions analogous to Equation (1)

$$\sum_{x_j \in \Gamma(x_k)} \eta_{kj}^M - \sum_{x_i \in \Gamma^{-1}(x_k)} \eta_{ik}^M = 0, \; x_k \in V_{joint}, \; M = I, II, III. \tag{2}$$

The challenge is to find vectors $\eta^M$, $M = I, II, III$. Instead of values $\eta_{ij}^M$, mass concentrations $r_{ij}^M$ can be considered using the relation (in practice, mass, molar, and volumetric concentrations are used, without difficulty they can be mutually converted. In this paper, mass concentrations are used, since for them material balance equations are most naturally written).

$$\eta_{ij}^M = r_{ij}^M \xi_{ij}, \ (i,j) \in E, \ M = I, II, III. \tag{3}$$

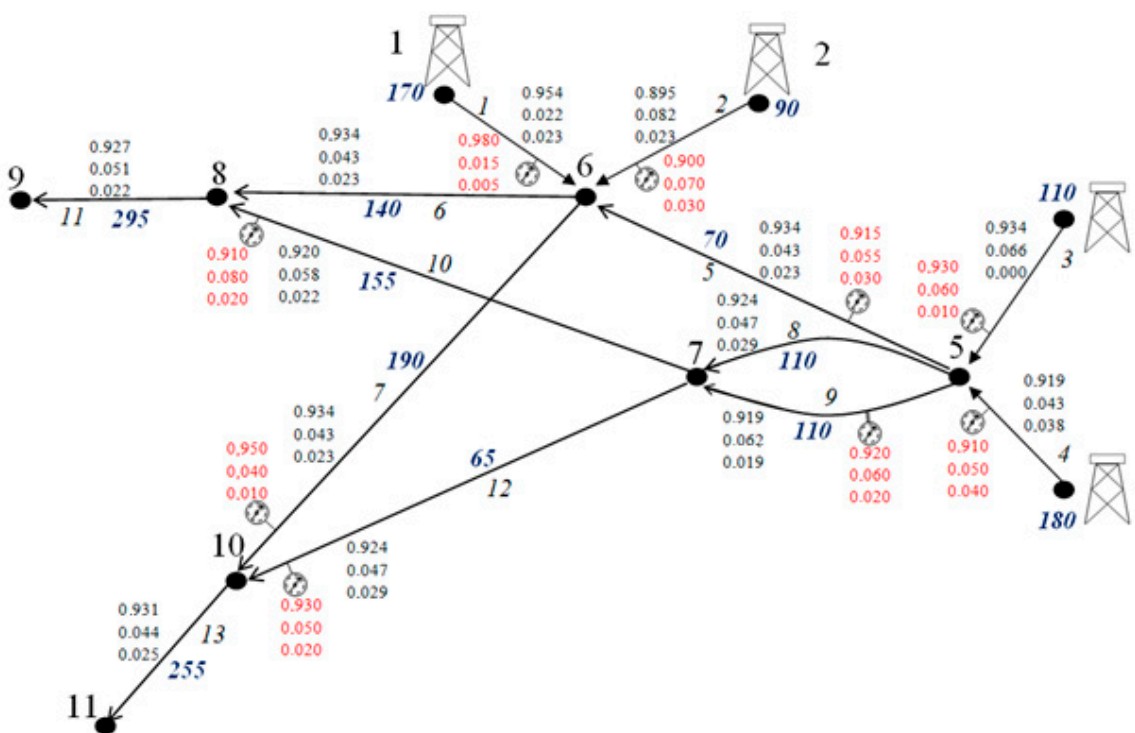

**Figure 1.** Gas transmission system (GTS) structure, concentration measurements at gas metering stations, calculation results. Designations: **1**—node numbers, *9*—arc numbers, 0.950—component concentration (measurement $r_{ij}^{*M}$), 0.950—component concentration (calculation $\hat{r}_{ij}^M$), *110*—flow rate, MSCMD (million standard cubic meters per day).

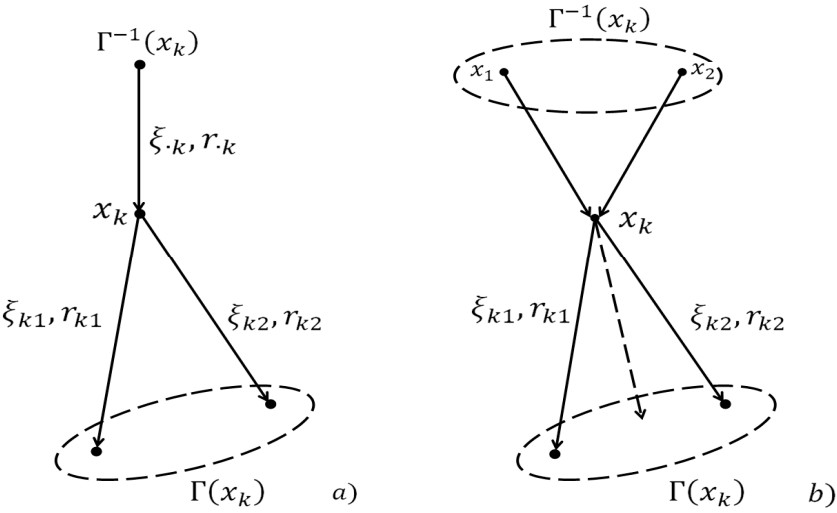

**Figure 2.** Schemes of separation (**a**) and merging (**b**) in the junction node $x_k$.

The solution to the problem considered in the paper, like any other problem of operational control, should be based on operational information: current measurements of the parameters of the technological process. The actual location of the GMS at the UGSS has been determined by historically significant factors, but not by the requirements for a reasonable assessment of the distribution of gas CC along transportation directions. In order to increase the degree of reliability of information on distribution of calorific value and/or makeup of gas in the system, the total of measurements at

UGSS facilities should be accounted for. This approach makes allowance for interdependence of the measured operating parameters and requires special methods, adequate mathematical and computer models to be implemented.

In the process of transportation, the concentrations change with the mixing of fluids, at the junction points of the pipelines. Along the length of each pipeline, the concentration does not change, so it is natural to associate it with the pipeline, regardless of where the measuring point is located. The problem is to estimate the concentration values for each pipeline based on the set of measurements. Measurements are erroneous, which means that the measured concentration value is a random value. At some known points of the system (GMS locations) the fluid composition is measured. Issues of metrological data uncertainty and measurement errors at GTS are treated in [23–25].

The source of information for solving this problem is a set of composition measurements, that is, concentration measurements $r_{ij}^*$. The measured points are assigned to the arcs of the graph (this can be done since the concentration of each component at the beginning and end of any pipeline is the same); for the set of such arcs, we introduce the notation $E^*$ (in Figure 1 $E^* = \{1, 2, 3, 4, 5, 7, 9, 10, 12\}$). We will also use an asterisk for the measured concentrations. The measurement result consists of the true (but unknown) value and the measurement error $\left(r_{ij}^M\right)^* = r_{ij}^M + \delta r_{ij}^M$, $(i, j) \in E^*$, $M = I, II, III$. In the theory of errors, measurement results are considered to be normally distributed quantities:

$$\delta r_{ij}^M \in N\left(0, \left(\sigma_{ij}^2\right)^M\right), \ M = I, II, III. \tag{4}$$

Symbol $X \in N\left(a, \sigma^2\right)$ means that random variable $X$ has a normal distribution with the mathematical expectation $a$ and the dispersion $\sigma^2$. Dispersions $\sigma_{ij}^2$ characterize the device (or measurement method) error on the arc $(i, j) \in E$. Error $\delta\eta_{ij}^M$ in determining the flow rate of component $M$ also has a normal distribution $\delta\eta_{ij}^M \in N\left(0, \xi_{ij}^2\left(\sigma_{ij}^2\right)^M\right)$, $M = I, II, III$. According to the problem statement, it is required to find such component flow distribution that is most consistent with the entire set of concentration measurements.

The problem under consideration in a certain sense resembles the well-known problem of "unaccounted gas" [26–29]. This problem requires, on the basis of operating parameter measurements, to determine gas balance more precisely (injections in the system, deliveries to consumers, own use), find imbalance reasons, obtain estimates of the flow rates in the directions of pumping, which are most consistent with the current measurements and the dynamics of their change over the past period.

In the stochastic formulation of the problem, it is customary to speak not of determining unknown quantities $r_{ij}$, but of their estimation. Mathematical statistics for point estimation of an unknown parameter recommends use of the maximum likelihood method (MLM). The estimation of maximum likelihood is the value of the argument at which the likelihood function takes its maximum. The likelihood function in our case is the probability of the totality of all measured values. MLM brings us to the problem of quadratic function conditional minimization [6]:

$$\sum_{M=I}^{III} \sum_{(i,j)\in E^*} \left(\left(\eta_{ij}^{*M} - \eta_{ij}^M\right)/\xi_{ij}\sigma_{ij}^M\right)^2 \to min. \tag{5}$$

In fact, not flow rates are measured, but component concentrations. However, using Equation (3), "measurements" $\eta_{ij}^{*M}$ can be defined through concentration measurements $r_{ij}^{*M}$, since values $\xi_{ij}$ are given. The minimum of criterion (5) should be sought provided abidance by constraints in the form of equalities and inequalities. Of greatest interest are the restrictions in the form of inequalities arising from mixing process non-equilibrium conditions. These issues are discussed in the next section.

*2.3. Conditions for Non-Equilibrium of Mixing Process*

The processes of mixing natural gas components are non-equilibrium (see Section 2.1). The laws of conservation of mass impose restrictions on the concentrations of the components on the output lines of the joint nodes. The obvious limitations are inequalities:

$$\underline{r}_{in} \le \underline{r}_{out} \le \overline{r}_{out} \le \overline{r}_{in}. \tag{6}$$

The subscript *in* refers to the input lines, the subscript *out* index refers to the output lines, the bar at the top indicates the maximum concentration, the bar at the bottom means the minimum one. Using the notation in Figure 2b, the variables included in inequalities (6) for each component are written in the form $\underline{r}_{in}^M(x_k) = \min\limits_{x_i \in \Gamma^{-1}(x_k)} \left( r_{ik}^M \right)$, $\overline{r}_{in}^M = \max\limits_{x_i \in \Gamma^{-1}(x_k)} \left( r_{ik}^M \right)$, $\underline{r}_{out}^M(x_k) = \min\limits_{x_i \in \Gamma(x_k)} \left( r_{ki}^M \right)$, $\overline{r}_{in}^M = \max\limits_{x_i \in \Gamma(x_k)} \left( r_{ki}^M \right)$, $M = I, II, III$. Inequalities (6) are directly proved using algebraic calculations. Let us consider the joint node of pipelines with 2 input and 2 output lines (Figure 2b). Fluid flow rates at the input lines are $\xi_{1,in}$, $\xi_{2,in}$. Suppose that the flow through the 1st line is divided into 2 parts: $\xi_{1,in} = \alpha_1 \xi_{1,in} + (1 - \alpha_1)\xi_{1,in}$. The first of them went through the 1st output line, the second through the 2nd output line. In the same way the flow is divided in the second output line. Thus, the fluid flow rates at the output lines equal $\xi_{1,out} = \alpha_1 \xi_{1,in} + \alpha_2 \xi_{2,in}$ $\xi_{2,out} = (1 - \alpha_1)\xi_{1,in} + (1 - \alpha_2)\xi_{2,in}$ accordingly. Consider any component of the mixture. Its concentration in the 1st input line is denoted $r_1$, and that in the 2nd input line $r_2$. In our reasoning, one component stands out; all other mixture components play the role of the second component with respective concentrations $1 - r_1$, $1 - r_2$ in the incoming lines, so $\xi_{1,in} = r_1\xi_{1,in} + (1 - r_1)\xi_{1,in}$, $\xi_{2,in} = r_2\xi_{2,in} + (1 - r_2)\xi_{2,in}$. Fluid flow rate on the 1st outlet line will consist of 2 summands, each one corresponding to one of the two components $\xi_{1,out} = [\alpha_1 r_1\xi_{1,in} + \alpha_2 r_2\xi_{2,in}] + [\alpha_1(1 - r_1)\xi_{1,in} + \alpha_2(1 - r_2)\xi_{2,in}]$. The selected component concentration in the 1st output line equals:

$$r_{1,out} = (\alpha_1 r_1 \xi_{1,in} + \alpha_2 r_2 \xi_{2,in}) / (\alpha_1 \xi_{1,in} + \alpha_2 \xi_{2,in}) \tag{7}$$

If, for example, $r_1 < r_2$, that is $\underline{r}_{in} = r_1$, then from relation (7) we directly obtain $\underline{r}_{in} = r_1 < r_{1,out}$. In the same way, inequality $r_{1,out} \le \overline{r}_{in}$ and similar inequalities for $r_{2,out}$ are proved. Relations (6) are proved for the arbitrarily chosen mixture component; therefore, its record can be generalized as:

$$\underline{r}_{in}^M \le \underline{r}_{out}^M \le \overline{r}_{out}^M \le \overline{r}_{in}^M, \ M = I, II, III. \tag{8}$$

This reasoning carried out for the special case of two incoming and two outgoing lines can be applied to a joint node with an arbitrary number of incoming and outgoing lines. The general outline of the proof is preserved. Let us mentally divide the incoming flow along each line into jets, and we obtain an analogue of relation (7) for each outgoing line. Hence the required result follows. Relations (8) are *necessary*, but *not sufficient*. They establish inequalities for the extreme (maximum and minimum) output concentrations. But they do not account for relations of the mixing component quantities at the input and output of the node. Indeed, consider again a certain component and imagine that its maximum concentration on the output lines is equal to the maximum concentration on the input lines $\overline{r}_{out} = \overline{r}_{in}$. For these lines, according to the physics of the process, the flow rate of the fluid in the outlet line cannot exceed the flow rate of the fluid in the inlet line $\overline{\xi}_{out} \le \overline{\xi}_{in}$. Similar ratios should hold for two, three, etc. output lines.

Let us write down these relations for the general case. The main result is presented by relations (13). The general outline of the reasoning is the same as in the particular case (Figure 2b), but the details of the proof are omitted due to the cumbersomeness. We take a node, denote the number of input lines by $N_{in}$, and the number of output lines by $N_{out}$. For analysis, it is not enough to consider only lines with maximum and minimum concentrations. Arranging in decreasing order concentrations in the input lines and in the output lines, we obtain two non- increasing sequences:

$$r_{in}^{(1)} \geq r_{in}^{(2)} \geq \ldots \geq r_{in}^{(N_{in})}. \tag{9}$$

$$r_{out}^{(1)} \geq r_{out}^{(2)} \geq \ldots \geq r_{out}^{(N_{out})}. \tag{10}$$

The superscript in parentheses indicates the rank of the corresponding number in the sequence. It is obvious that $r_{in}^{(1)} = \bar{r}_{in}$, $r_{in}^{(N_{in})} = \underline{r}_{in}$ and $r_{out}^{(1)} = \bar{r}_{out}$, $r_{out}^{(N_{out})} = \underline{r}_{out}$. Using the elements of sequence (9), we construct a line of "limit concentrations". This will be a piecewise smooth curve $r(x) = r_{in}(x)$:

$$r(x) = \begin{cases} r_{in}^{(1)} \text{ provided } 0 \leq x \leq \xi_{in}^{(1)} \\[2mm] \dfrac{r_{in}^{(1)}\xi_{in}^{(1)} + r_{in}^{(2)}\left(x - \xi_{in}^{(1)}\right)}{x} \text{ provided } \xi_{in}^{(1)} < x \leq \xi_{in}^{(1)} + \xi_{in}^{(2)} \\[2mm] \ldots \\[2mm] \dfrac{r_{in}^{(N_{in}-1)}\sum_{j=1}^{N_{in}-1}\xi_{in}^{(j)} + r_{in}^{(N_{in})}\left(x - \sum_{j=1}^{N_{in}-1}\xi_{in}^{(j)}\right)}{x} \text{ provided } \sum_{j=1}^{N_{in}-1}\xi_{in}^{(j)} < x \leq \sum_{j=1}^{N_{in}}\xi_{in}^{(j)} \end{cases} \tag{11}$$

Let us proceed to give an interpretation of the point with coordinates $[x; r(x)]$ and take by way of example a point lying in the 3rd half-segment of its definition $\xi_{in}^{(1)} + \xi_{in}^{(2)} < x \leq \xi_{in}^{(1)} + \xi_{in}^{(2)} + \xi_{in}^{(3)}$. The inlet flow rate maximum concentration equal to $x$ is obtained if we sum the total gas supply through the inlet lines with concentrations $r_{in}^{(1)}$ and $r_{in}^{(2)}$ and with partial supply through the line with concentration $r_{in}^{(3)}$, provided flow rate $x - \left(\xi_{in}^{(1)} + \xi_{in}^{(2)}\right)$. The maximum concentration for flow rate $x$ is equal to $(1/x)\left[r_{in}^{(1)}\xi_{in}^{(1)} + r_{in}^{(2)}\xi_{in}^{(2)} + r_{in}^{(3)}\left(x - \xi_{in}^{(1)} - \xi_{in}^{(2)}\right)\right]$.

The distribution of flow rates and concentrations in the output lines will be acceptable if all points with coordinates:

$$\left[\xi_{out}^{(1)}; r_{out}^{(1)}\right], \left[\xi_{out}^{(1)} + \xi_{out}^{(2)}; \frac{r_{out}^{(1)}\xi_{out}^{(1)} + r_{out}^{(2)}\xi_{out}^{(2)}}{\xi_{out}^{(1)} + \xi_{out}^{(2)}}\right], \ldots, \left[\sum_{j=1}^{N_{out}}\xi_{out}^{(j)}; \frac{\sum_{j=1}^{N_{out}} r_{out}^{(j)}\xi_{out}^{(j)}}{\sum_{j=1}^{N_{out}}\xi_{out}^{(j)}}\right] \tag{12}$$

lie no higher than curve (11). The first of these points corresponds to the output line with maximum concentration $r_{out}^{(1)}$, the 2nd one to two output lines with concentrations $r_{out}^{(1)}$ and $r_{out}^{(2)}$, etc. The last point with abscissa $\xi_{\Sigma} = \sum_{j=1}^{N_{out}}\xi_{out}^{(j)}$ and ordinate $\rho$ always lies on the curve $r(x)$. Here $\rho$ is the system average value of the considered component. If the remaining points are no higher than $r(x)$, then the concentration distribution (10) is acceptable. If for ordinates of points in the aggregate (9) we introduce the notation $y^{(1)} = r_{out}^{(1)}$, $y^{(2)} = \frac{r_{out}^{(1)}\xi_{out}^{(1)} + r_{out}^{(2)}\xi_{out}^{(2)}}{\xi_{out}^{(1)} + \xi_{out}^{(2)}}, \ldots, y^{(N_{out})} = \frac{\sum_{j=1}^{N_{out}} r_{out}^{(j)}\xi_{out}^{(j)}}{\sum_{j=1}^{N_{out}}\xi_{out}^{(j)}}$, then the admissibility condition is written in the form:

$$y^{(1)} \leq r\left(\xi_{out}^{(1)}\right), \ y^{(2)} \leq r\left(\xi_{out}^{(1)} + \xi_{out}^{(2)}\right), \ldots, y^{(N_{out})} \leq r\left(\sum_{j=1}^{N_{out}}\xi_{out}^{(j)}\right). \tag{13}$$

This condition (13) is necessary and sufficient for non-equilibrium process feasibility. It should be noted that the verification procedure (9)–(13) must be carried out for each fluid component.

Now we can fully formulate the problem of estimation of CC gas mixture distribution in the directions of aggregated scheme transportation, listing all the restrictions that should be accounted for in the optimization problem with criterion function (5).

Constraints in the form of equalities include balance conditions in the joint nodes for each component:

$$\sum_{x_j \in \Gamma(x_k)} \eta_{kj}^{M} - \sum_{x_i \in \Gamma^{-1}(x_k)} \eta_{ik}^{M} = 0, \ x_k \in V_{joint}, \ M = I, II, III. \tag{14}$$

Conditions of equality of the fluid flow rate on each arc to the sum of all components flow rates on the arc:

$$\sum_{M=I}^{III} \eta_{ij}^M = \xi_{ij}, \ (i,j) \in E. \tag{15}$$

Moving to concentrations, we write down this equation in the form:

$$\sum_{M=I}^{III} r_{ij}^M = 1, \ (i,j) \in E. \tag{16}$$

Limitations in the form of inequalities include: condition (13) of a non-equilibrium process feasibility, which must be satisfied for each fluid component $M = I, II, III$; non-negativity of all concentrations of the problem:

$$r_{ik}^M \geq 0, \ (i,k) \in E, \ M = I, II, III. \tag{17}$$

Thus, determination of fluid composition per transportation direction is reduced to optimization problem (5) under constraints (14), (15) in the form of equalities and constraints (13), (17) in the form of inequalities.

### 2.4. Assessment of the Information Content of the Gas Transmission System (GTS) Metrological Equipment

Before proceeding with solution of the task of estimating concentrations $r_{ik}^M$, it is necessary to find out what can be obtained from the available totality of measurements $r_{ik}^{*M}$, $(i,k) \in E^*, M = I, II, III$. Estimates obtained by solving the problem of conditional minimization will be marked with the caret, for example, concentration estimates we shall notate as $\hat{r}_{ik}^M$. The estimation quality depends on the quantity and location of the instruments, instruments accuracy class and the frequency of instruments polling sessions. With an insufficient number of measurements it is impossible to obtain all estimates of all concentrations $\hat{r}_{ik}^M$, $(i,k) \in E$. To demonstrate the possibilities of metrological support of the system, let us consider the flow of a two-component mixture on the simplest system - a tee with one input line and two output lines (Figure 2a). By $n^*$, $m_{in}$, $m_{out}$, $m_{joint}$ we denote the number of measuring points, sources, outflow and joint nodes, respectively.

In the beginning we analyze the simplified problem (5), (14). It is a quadratic program with $n$ unknowns and $m_{joint}$ linear constraints. Different cases may arise depending on values $n^*$, $m_{in}$, $m_{out}$, $m_{joint}$. For a three-way piece (Figure 2a) $n = 3$, $m_{joint} = 1$ there can be 3 options of measured parameters, respectively $n^* = 3, 2, 1$. In option 1 (concentrations are measured in each pipeline) to estimate unknowns $r_{ij}$, we resort to the problem of minimizing quadratic function of 3 variables with one linear constraint. Using the constraint, we express one of the unknown quantities $r_{ij}$ in terms of the other two. The resulting problem of unconditional extremum of 2 variable quadratic function can be reduced to a system of 2 linear equations with the same number of unknowns. Solving this problem, we obtain estimates $\hat{r}_{ij}$ of unknowns $r_{ij}$ which use the results of all 3 measurements. The estimates $\hat{r}_{ij}$ are better substantiated than each value direct measurements $r_{ij}^*$. In option 2, the concentration is measured only in 2 lines (for example, $r_{k1}^*, r_{k2}^*$ in Figure 2a). Now $n = 3$, $n^* = 2$, $m_{joint} = 1$, we have the problem of minimizing quadratic functions of 2 variables $r_{k1}, r_{k2}$, provided one linear constraint. The maximum of the likelihood function is found directly, and the estimates are equal to measurements $\hat{r}_{k1} = r_{k1}^*$, $\hat{r}_{k2} = r_{k2}^*$. The constraint is used to find an estimate of the missing unknown concentration along the arc entering the node $x_k$. In option 3, only one value is measured, for instance $r_{k1}^*$ (Figure 2a). The maximum likelihood method allows one to obtain only a trivial result—an estimate of the concentration along that arc $\hat{r}_{k1} = r_{k1}^*$.

The same kind of reasoning is carried out in the general case for any ratio of quantities $n$, $n^*$, $m_{joint}$. It helps to reveal what results can in principle be obtained with the existing system of measuring gas composition. Concentration estimates cannot always be obtained and not for all graph edges. It all

depends on the number and location of measuring points. For the graph of arbitrary configuration, of greatest importance is the value $d = m_{joint} - (n - n^*)$. When $d > 0$, sought estimates can be refined by accounting for mutual influence of all the measurements. If $d = 0$, then the measurement results directly are concentration estimates. If $d < 0$, then the constraints in the form of equalities are insufficient to estimate all unmeasured concentrations $r_{ij}$. In the case $d \leq 0$ graph $G$ can contain subgraphs that meet the condition $d > 0$. (Subgraph $G_l(X_l, E_l)$ of the graph $G(X, E)$ is a graph for which $X_l \subset X$ and for each graph node $x_k \in X_l$, $\Gamma_l(x_k) = \Gamma(x_k) \cap X_l$, [21,22]). Consequently, sufficient measurements are made on the arcs of these subgraphs to refine concentration estimates. An algorithmic procedure for identifying such subgraphs has been developed. Estimates are obtained as linear functions of measurements

$$\hat{r}_{ij} = \sum_{(k,l) \in E^*} \left[ a_0^{ij} + a_{kl}^{ij} r_{kl}^* \right], \ (i, j) \in E.$$

The quality of the estimates is characterized by their dispersion. Under assumption (4), dispersion of the estimates resulting from the maximum likelihood method is calculated as

$$\mathbf{D}\hat{r}_{ij} = \sum_{(k,l) \in E^*} \left[ \left( a_{kl}^{ij} \right)^2 \mathbf{D} r_{kl}^* \right], \ (i, j) \in E.$$

### 2.5. Computational Aspects

The considered problem of the distribution of the CC of multicomponent fluids is thus reduced to a mathematical programming problem with constraints in the form of equalities and inequalities. The number of unknowns and constraints in a task can be large. When solving it numerically, pitfalls can be encountered, the manifestation of which will be the flatness or, conversely, the ravine of the target function. The idea of the presence or absence of these phenomena, negative from a computational point of view, can be obtained by conducting a computational experiment. We conducted experiments, and found non-standard computational techniques and algorithms, which we consider to be proof of the efficiency of the developed technique.

### A. Two Approaches to Formalizing the Problem

The first approach (unknown variables–flow rates or concentration of components, see Sections 2.2–2.5): the sought variables are concentrations $r_{ij}^M$. If in relations (5), (14) and 15), instead of component flow rates $\eta_{ij}^M$, using relations (3), we substitute their expressions in terms of concentrations $r_{ij}^M$, then we will obtain a formalization of the problem of determining gas composition in variables $r_{ij}^M$ (component concentrations on the arcs of calculated graph) or the problem of optimizing quadratic function with constraints in the form of equalities and inequalities. Since the fluid flow rates $\xi_{ij}$ are considered known, we can act differently, taking values $\eta_{ij}^{*M} = \xi_{ij} r_{ij}^{*M}$ as initial information for calculation.

The second approach: the sought variables are coefficients $\alpha_{ik}^j$. We introduce new unknowns $\alpha_{ik}^j$, presenting the fluid flow rates in the form

$$\xi_{kj} = \sum_{i \in \Gamma^{-1}(x_k)} \alpha_{ik}^j \xi_{ik}, \ x_k \in V_{joint}. \tag{18}$$

Here, we used the notation in Figure 2b. The physical meaning of relation (18) is as follows. We consider joint node $x_k \in V_{joint}$. Each flow through the incoming line $(i, k)$ (with flow rate $\xi_{ik}$) is divided into "jets" $\alpha_{ik}^1 \xi_{ik}, \alpha_{ik}^2 \xi_{ik}, \ldots, \alpha_{ik}^{N(k)} \xi_{ik}$; $N(k) = mes\ \Gamma(x_k)$, where $\alpha_{ik}^j$ is the fraction of the flow rate $\xi_{ik}$ entering the output line $(k, j)$, $mes\ \Gamma(x_k)$ is the number of arcs outgoing from node $x_k$. Obviously, for $\alpha_{ik}^j$ the following relations are true:

$$\sum_{j=1}^{N(k)} \alpha_{ik}^j = 1, x_i \in \Gamma^{-1}(x_k); \ \alpha_{ik}^j \geq 0. \tag{19}$$

In accordance with relation (18), we obtain the distribution of the components in the outgoing lines:

$$\eta_{kj}^M = \sum_{i \in \Gamma^{-1}(x_k)} \alpha_{ik}^j \eta_{ik}^{j,M}, M = I, II, III, \ x_k \in V_{joint}. \tag{20}$$

An obvious fact is used in relations (19). Let us consider the fluid flow along any incoming line. In the joint node, this flow is divided into parts ("jets") along the outgoing lines ($N(k)$ is the number of the jets). Component composition of each of these parts will be the same, the fact on which formula (20) is based. Thus, if values $\eta_{ik}^M, M = I, II, III$, are taken as unknowns, then the mathematical model does not require inclusion of inequality type relations (13), following from the conditions of non-equilibrium of the process. These relations will be performed automatically.

The technique of mentally dividing each stream entering the node into "jets" was used above (see Section 2.3, Formula (7)). If values $\alpha_{ik}^j$ are determined, then the known concentrations are used to calculate the concentrations on all network arcs. This can be done beginning with concentrations in the sources $r_{ij}^{*M}$, $x_i \in V_{in}$, $x_j \in \Gamma(x_i)$, $M = I, II, III$. The concentrations are calculated sequentially in the order determined by the following node numbering algorithm.

*Algorithm.* We number the sources (in random order) with the numbers $1, 2, \ldots, N_{in}$. By numbering a node, we color it and all arcs outgoing from it. We assign the next number to the node, all incoming arcs in which are colored. We continue the procedure until all nodes are numbered.

In the same way, all flow rates $\eta_{ij}^M$, $(i, j) \in E$, $M = I, II, III$ can be expressed through a set of quantities $\alpha_{ik}^j$, which we designate as vector $\boldsymbol{\alpha} = \|\alpha_{ik}^j\|$. In particular, we obtain measured values $\eta_{ij}^{*M} = \eta_{ij}^{*M}(\boldsymbol{\alpha})$, $(i, j) \in E^*$, $M = I, II, III$. As a result, the problem is reduced to finding the minimum of criterion function (5) subject to fulfillment of constraints (15) and (19). Note that the calculation procedure automatically satisfies conditions $\eta_{ik}^M \geq 0$, $(i, k) \in E$, $M = I, II, III$, and, most importantly, conditions (13) of the mixing process non-equilibrium. The last statement is explained by the fact that when deriving non-equilibrium conditions (13), the method of splitting the flow along each incoming line into "jets" was used (see Section 2.3, derivation of formulae (7), (8)).

In the first and second approaches, either component flow rates $\eta_{ij}^M$ or their concentrations $r_{ij}^M$ are used. By virtue of relations (3), these variables are easily expressed through each other. However, the choice of variables can influence the effectiveness of the calculation methods.

*B. The First Approach. Calculation Procedure Specificity*

Nested solutions method. In its complete form, the problem of mathematical programming (5) with conditions (13)–(16) because of a large number of restrictions and significant difference in the sought variable values can be computationally difficult. In this case, one can resort to the method of finding solution to the problem disregarding some constraints. The obtained quasi-solution should be further checked for compliance with previously ignored constraints. If it does comply, then it is the desired result. If it fails to comply, then those of the ignored constraints that are not satisfied in the quasi-solution are added to the conditions. For example, constraints (13) are rather cumbersome. Instead of them, restrictions (8) can be introduced, that is, only part of restrictions (13). A quasi-solution with constraints (8) can satisfy all constraints (13), then a solution to the problem is reached.

The computational experiment performed proved the method to be very effective, although, of course, it is impossible to guarantee its success in any situation. The possibility of rejecting condition (17) was also tested. In the experiments, after the quasi-solution was obtained, condition $r_{ik}^M \geq 0$ was introduced only for those unknown concentrations that turned out to be negative in the quasi-solution. In the solution obtained in this case, all constraints were satisfied.

The larger the dimension of the problem, the more efficient the application of the nested solution method. The application of the method in computer systems for operational control is especially justified. For operational control, it is characteristic that with the next arrival of information about measurements from the GMS, a good initial approximation is usually known to solve the problem of

determining the component composition. Therefore, first of all, it is necessary to take into account the constraints in the form of inequalities, which are currently fulfilled as equality.

## 3. Results

Section 3 uses a double designation of arcs, for example, *1* and (1, 6) (see Figure 1), the 2nd way is more convenient in those cases when it is necessary to emphasize the direction of the arc.

### 3.1. Example

To illustrate the technique developed, we will give an example of calculating CC distribution in a three-component mixture flow in the graph shown in Figure 1. Structural graph *G* contains 13 arcs (transportation directions), 11 nodes (of which 4 sources, 2 outflows, 5 joints). Measurement points are located on 9 arcs. The criterion for the information content is $d = m_{joint} - (n - n^*) = 5 - (13 - 9) = 1$. Therefore, in this case the technique provides an opportunity to take into account mutual influence of the measurements. The initial measurement data and optimization problem solution results are shown in Table 1 and duplicated in Figure 1. The values of the estimates $\hat{r}_{ij}^{M}$ of the component concentrations shown in Figure 1, as well as in Table 1 are obtained by computer calculations and rounded to 3 decimal places. Naturally, the estimates $\hat{r}_{ij}^{M}$ were obtained for all arcs, including those without measuring devices (arcs 6, 8, 11, 13, Figure 1). Due to rounding errors, relation (16) for estimates $\hat{r}_{ij}^{M}$ may be violated in the last digit. So, for the arc (1, 6) we have $\sum_{M=I}^{III} \hat{r}_{16}^{M} = 0.924 + 0.022 + 0.023 = 0.999$.

**Table 1.** Composition measurements $r_{ij}^{*M}$ and estimates $\hat{r}_{ij}^{M}$ (numerator—measurement, denominator—calculation).

| Arc Nº/(i, j) | Flow Rate, MSCMD | Concentration of Components | | |
|---|---|---|---|---|
| | | M = I | M = II | M = III |
| 1/(1, 6) | 170 | 0.980/0.954 | 0.015/0.022 | 0.005/0.023 |
| 2/(2, 6) | 90 | 0.900/0.895 | 0.070/0.082 | 0.030/0.023 |
| 3/(3, 5) | 110 | 0.930/0.934 | 0.060/0.066 | 0.010/0.000 |
| 4/(4, 5) | 180 | 0.910/0.919 | 0.050/0.043 | 0.040/0.038 |
| 5/(5, 6) | 70 | 0.915/0.934 | 0.055/0.043 | 0.030/0.023 |
| 6/(6, 8) | 140 | —/0.934 | —/0.043 | —/0.023 |
| 7/(6, 10) | 190 | 0.950/0.934 | 0.040/0.043 | 0.010/0.023 |
| 8/(5, 7) | 110 | —/0.924 | —/0.047 | —/0.029 |
| 9/(5, 7) | 110 | 0.920/0.919 | 0.060/0.062 | 0.020/0.019 |
| 10/(7, 8) | 155 | 0.910/0.920 | 0.080/0.058 | 0.010/0.022 |
| 11/(8, 9) | 295 | —/0.927 | —/0.051 | —/0.022 |
| 12/(7, 10) | 65 | 0.930/0.924 | 0,050/0,047 | 0.020/0.029 |
| 13/(10, 11) | 255 | —/0.931 | —/0.044 | —/0.025 |

The difference between direct measurements $r_{ij}^{*M}$ and estimates $\hat{r}_{ij}^{M}$ for the entire set of measurements, especially for components with low concentrations, in some cases is very large. For example, $r_{(7,8)}^{*II} = 0.080$, $\hat{r}_{(7,8)}^{II} = 0.058$, $r_{1,6}^{*III} = 0.005$, $\hat{r}_{1,6}^{III} = 0.023$, $r_{(3,5)}^{*III} = 0.010$, $\hat{r}_{(3,5)}^{III} = 0.000$. Significant discrepancies in the values $r_{ij}^{*M}$ and $\hat{r}_{ij}^{M}$ may indicate non-fulfillment of relation (4), in the part that refers to the absence of systematic measurement errors. The presence of significant discrepancies should be an incentive for the standardization of measuring instruments.

The distribution of flows ξ is included in the initial data. The flow rates of the fluid in the directions of pumping $\xi_{(1,6)} = 170, \xi_{(2,6)} = 90, \ldots$ are balanced, and they are shown in Figure 1 and duplicated in Table 1.

Despite the small size of the graph, it allows you to illustrate the characteristic features of the problem of assessing the CC of the GTS pipelines. The sample data have been selected to illustrate the methods.

The concentration measurement data is deliberately selected in such a way as to simulate possible errors. For example, the measurements of the concentration of component $M = II$ on arcs emanating from the sources are 0.015, 0.070, 0.060, 0.050, while the concentration $M = II$ measurement on the outgoing arc (7, 8) is 0.080. Naturally, such a situation is impossible for true concentration values due to violation of the non-equilibrium conditions. Nevertheless, the solution of the problem has been obtained, and the calculation results satisfy the conditions of non-equilibrium in both soft (8) and harder (13) forms. This means that the algorithm made it possible to smooth out the measurement errors without going beyond the limits, including (16) and (17).

Attention should be paid to the difference in the component composition of the fluid along parallel lines 8 and 9 (arcs between nodes 5 and 7). Thus, $\hat{r}_8^{II} = 0.047$, $\hat{r}_9^{II} = 0.062$, $\hat{r}_8^{III} = 0.029$, $\hat{r}_9^{III} = 0.019$. This once again indicates that the distribution of the component composition of the fluid is influenced by a set of interrelationships of operating parameters throughout the GTS. Rough-and-ready, local methods for solving the problem can lead to significant errors.

One of the purposes of this example is to test nested decision methods (see Section 1, Section 2.5B). First, the optimization problem was solved without taking into account inequality constraints. In this setting, the problem is reduced to a quadratic program with linear equality constraints. The resulting solution contained 2 negative components and one violated inequality (8). Then, both inequalities from (17) and the violated inequality from (8), which were not fulfilled at the first stage, were introduced into the model. The solution to this corrected problem has satisfied all the constraints.

It is interesting to note that none of the constraints (13), which are not included in the number of inequality constraints (8), were violated either at the 1st or at the 2nd stage of the solution. Apparently, restrictions from (8) are infrequently violated, which indicates the usefulness of the proposed nested solution method.

In this example, both approaches to the formalization of the problem $A$ and $B$ (Section 2.5) are tested. They have been proven capable of working, both lead to virtually identical numerical results.

Comparison of approaches A and B can be, in our opinion, carried out only on the basis of a computational experiment. However, it is clearly premature to draw conclusions based on the study of only one example. Our assumption about the preferability of method B remains a hypothesis that must be verified by a representative computational experiment.

### 3.2. Mixing Processes Non-Equilibrium Indicator

Initially, the study of irreversible and non-equilibrium processes was the subject of thermodynamics [17–19]. Then, other branches of science that lie at the intersection of physics and chemistry arose to study processes of this kind (see Section 2.1). In connection with the need to determine the directionality of the physical (heat transfer) process the concept of entropy was introduced, which underlies one of the cardinal laws of physical laws—the second law of thermodynamics. Entropy is a physical quantity deduced from mathematical models of energy transfer processes. Entropy is a monotonic function defined in such a way that in irreversible heat transfer processes it inevitably increases. Currently, there are numerous options for extending the concept of entropy to processes not associated with heat transfer. K. Shannon [30] introduced the concept of information entropy as a measure of information, as a characteristic of the amount of information contained in a message. We can say that entropy is a general name for the quantitative characteristics of a wide variety of processes, primarily processes that are irreversible and characterised by non-equilibrium. For the processes of gas mixing in industrial pipelines considered here, the quantitative indicator of non-equilibrium can also be introduced.

To define it, we introduce the concept of an extended graph $G'$. $G'$ is obtained from the original graph $G$ by adding to it a fictitious source $s'$ and a fictitious outflow $t'$, as well as fictitious arcs connecting $s'$ to the sources of graph $G$ and outflows of the graph $G$ to $t'$. Fictitious arcs are shown in Figure 3 with a dotted line.

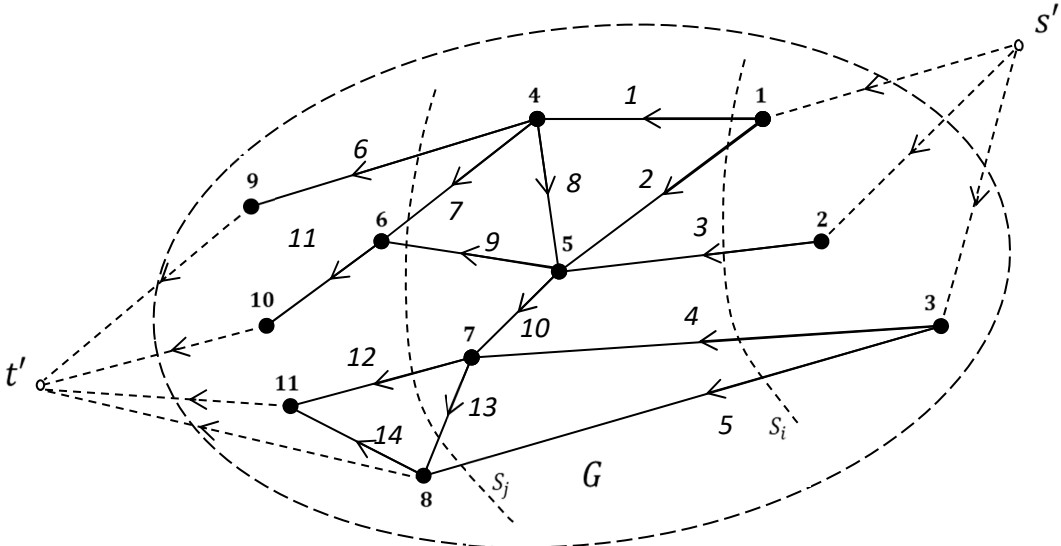

**Figure 3.** Cuts of the extended graph $G'$.

In graph $G'$ cuts separating $s'$ from $t'$ are considered. A minimal set of arcs, the removal of which splits the graph into 2 components of connectivity, of which one contains $s'$ and the other $t'$ is called a cut (more precisely, a minimal cut) separating the source $s'$ from the outflow $t'$ [21,22]. The term minimal cut means that when any arc is removed from the set, it ceases to be a cut.

Figure 3 shows 2 cuts $S_i$, $S_j$. The fluid flow rate distribution through the cut is determined by population $\alpha_{kj} = \xi_{kj}/\xi_\Sigma$, $(k,j) \in S_i$. Here $\xi_\Sigma = \sum\limits_{(k,j) \in S_i} \xi_{kj}$ is the total flow rate through the cut. All flow rates $\xi_\Sigma$ are obviously the same in any cut. Since $\sum\limits_{(k,j) \in S_i} \alpha_{kj} = 1$ it can be argued that the set $\alpha_{kj}, r_{kj}^M$, $(k,j) \in S_i$ for each component $M$ determines a discrete random variable specified in the cut $S_i$. The dispersion of this variable:

$$H^M(S_i) = \sum_{(k,j) \in S_i} \left( r_{kj}^M - \bar{r}_M \right)^2 \alpha_{kj}. \tag{21}$$

characterizes the degree of component $M$ scattering around the mean value $\bar{r}^M$, that is, degree of the component flow proximity to the state of complete mixing. The function of the cut:

$$H(S_i) = \sum_{M=I}^{III} H^M(S_i) \tag{22}$$

characterizes the mixture proximity to the equilibrium state, the state of complete mixing. Here, as before, it is assumed that the number of mixture components is *III*. According to established tradition, entropy as a characteristic of irreversible processes can change in only one direction: it can increase. Let us introduce the concept of "entropy" for the process of non-equilibrium mixing of multicomponent gas mixtures moving through industrial pipeline systems, trying not to break this tradition. Let us define the "entropy" of the cut $S_i$ using the formula:

$$\text{entr}(S_i) = -H(S_i) = -\sum_{M=I}^{III} H^M(S_i). \tag{23}$$

According to definition (23), the gas mixture "entropy" is the sum of the gas mixture component "entropies". In contrast to the overwhelming majority of other applications function $\text{entr}(S_i)$ is defined not for a continuum set, but for a discrete set: the set of cuts of the extended graph $G'$ of pipeline system.

Let us introduce a new concept. Each cut $S_i$ divides extended graph $G'$ into 2 parts, into 2 connected components, subgraphs $G'_{s'}(S_i), G'_{t'}(S_i)$ containing a source and an outflow, respectively, so that $G' = G'_{s'}(S_i) \cup G'_{t'}(S_i) \cup S_i$. We can say that cut $S_j$ includes $S_i$, and denote it as $S_j > S_i$ if $G'_{s'}(S_i) \subset G'_{t'}(S_j)$ (see Figure 3). The necessary condition for the non-equilibrium of component $M$ flow distribution in graph $G'$ is written in the form $H^M(S_i) > H^M(S_j)$.

Let us call the maximum number of arcs in the chain from $s'$ to $i$ as the distance from $s'$ to node $i$ and introduce a sequence of cuts $S_k$, $k \geq 1$. $S_k$ is a cut that divides extended graph $G'$ into 2 subgraphs, one of which $G'_{s'}(S_k)$ contains all the nodes removed from $s'$ at a distance not exceeding $k$, and the other subgraph $G'_{t'}(S_k)$ contains the remaining nodes of the graph. By construction, $S_j > S_k$, if $j > k$. We denote the distance from $s'$ to $t'$ through $K$ and consider a sequence of cuts $S_1 \prec S_2 \prec \ldots \prec S_K$. The function (22) $H(z)$ is defined for the set of cuts $S_1, S_2, \ldots, S_K$, that is, argument $z$ can be considered a real variable taking values $1, 2, \ldots, K$, or a real $n$-dimensional vector $\mathbf{z} \in R^n$ which values are determined by the cuts $S_1, S_2, \ldots, S_K$. We can consider, for example, $z_{jk} = \begin{cases} 1, & \text{if } (j,k) \in S_i \\ 0, & \text{if } (j,k) \notin S_i \end{cases}$, $(j,k) \in E$.

The above operation $S_j \prec S_i$ establishes a partial ordering relation in the set of cuts $S_1, S_2, \ldots, S_K$. Cuts $S_i, S_j$ in Figure 3 are related via ratio $S_j \prec S_i$. The "entropy" in these cuts satisfies the inequality $\text{entr}(S_i) \leq \text{entr}(S_j)$.

Table 2 shows values of indicator $\text{entr}(S_i)$ and its additive components $H^M(S_i)$ for 3 cuts of graph $G'$ (Figure 4).

**Table 2.** Example. Indicator $\text{entr}(S_i)$ values for 3 cuts (Figure 4).

| Cut $S_i$ | Arcs of the Cut | $\text{entr}(S_i)$ | $H^I(S_i)$ | $H^{II}(S_i)$ | $H^{III}(S_i)$ |
|---|---|---|---|---|---|
| $S_1$ | 1, 2, 3, 4 | $-1.0800 \times 10^{-3}$ | $0.431 \times 10^{-3}$ | $0.468 \times 10^{-3}$ | $0.180 \times 10^{-3}$ |
| $S_2$ | 6, 7, 10, 12 | $-0.0823 \times 10^{-3}$ | $0.370 \times 10^{-4}$ | $0.415 \times 10^{-4}$ | $0.380 \times 10^{-5}$ |
| $S_3$ | 11, 13 | $-0.0170 \times 10^{-3}$ | $0.500 \times 10^{-5}$ | $0.110 \times 10^{-4}$ | $0.110 \times 10^{-5}$ |

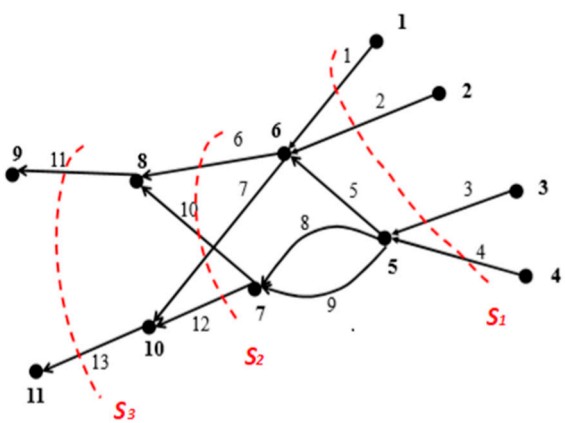

**Figure 4.** Example. The cuts $S_1, S_2, S_3$ (see the Table 2).

The indicator of non-equilibrium allows us to gain a general picture of the distribution of CC/caloric content in the GTS. This will help in solving the issues of adjusting the aggregated scheme in the direction of refinement or dividing the graph into subgraphs $G = G_u \cup G_v$, for one of which $G_u$ the non-equilibrium of the mixing process is essential, and for the other $G_v$ it is not essential. In this case, it may be acceptable the switch from the study of CC distribution on $G$ to study on $G_u$. "Entropy" rapidly decreases with distance from sources and approaching sinks. So $\text{entr}(S_1) : \text{entr}(S_2) = 13.1$, $\text{entr}(S_2) : \text{entr}(S_3) = 4.84$. This indicates a rapid approach of the process to the equilibrium state.

In general, the example testifies to the efficiency of the methods, the efficiency of computational procedures and the acceptability of the apparatus for practical calculations.

## 4. Conclusions

A technique is proposed that allows calculating the distribution of the component composition of natural gas flows through gas transportation systems of arbitrary configuration. A mathematical model has been developed that takes into account the irreversibility and non-equilibrium of mixing processes, as well as the random nature (instruments errors) of the component concentration measurement. Non-equilibrium conditions are derived for a mixture with an arbitrary number of components. The model takes into account the entire set of measurements and their mode-technological relationships. The possibility of two approaches to the numerical solution of the problem is established. The aim of the research is achieved by solving a mathematical programming problem with constraints in the form of equalities and inequalities. A subject-oriented computational procedure has been developed, which allows one to obtain a result using standard software packages. The efficiency of the technique is demonstrated by a numerical example of calculating the flow distribution of a three-component mixture. A non-equilibrium indicator («entropy» of the mixing process) is introduced, which is defined for a discrete set: the set of cuts of the computational graph.

**Funding:** This research received no external funding.

**Conflicts of Interest:** The author declares no conflict of interest.

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
