# Peer review of "Fluid Mixing Nonequilibrium Processes in Industrial Piping Flows"

_energies, doi:10.3390/en13236364_

Round 1

Reviewer 1 Report

The paper is interesting and concerns fluid mixing nonequilibrium processes in industrial piping flows. The paper is harmoniously structured between its parts. However, a few suggestions will surely improve the reading quality.

*) Although the abstract describes the content of the paper, it does not provide a summary of the most significant numerical results. Please supplement the abstract with a few sentences that fill this gap.

*) The introduction, although exhaustive as regards the specific methodology adopted, does not provide an overview of the mixture processes. It would be interesting to mention some theoretical-applicative fields in which such processes are highly acknowledged. For example, in the modeling of magneto-rheological fluids it would be interesting to quote the paper

doi: 10.1016/j.ijnonlinmec.2019.103288

as well as in multi-modal techniques as studied in

doi: 10.1002/qre.2458

doi: 10.1109/ICDMW.2011.135 

*) Furthermore, the most important findings are not detailed in the introduction as well as the structure of the paper.

*) Maybe, in Task formalization, the forma definition of the graph is due. Many definitions are reported discursively. Perhaps more formalism would increase the quality of the paper.

*) There are so many footnotes. Perhaps the MDPI publishing house does not allow their use, except in exceptional cases.

*) Please read Section 3.3 carefully. Its reading appears difficult as the text is poorly structured. A slight modification of the text would certainly help the reader to better understand the content of this important section.

Author Response

Added or corrected text fragments are highlighted in green in the manuscript. Fragments to be deleted are highlighted in red. Line numbers in column 2 are retained for reviewers' comments. The line numbers in column 4 are from the revised version of the manuscript.

Author Response

(The authors gave the same response as above.)

Reviewer 3 Report

Overall the work presents an adequate approach and analyzes an important matter. Several issues significantly lower manuscript quality.

Use of references in the abstract is not very useful.

Stating more clearly the actual progress beyond previous publications is needed.

The introduction does not provide a comprehensive view of the state-of-the-art. In its present form it is more of a technical report rather than a scientific article.

Model description is very detailed (even if most of the equations are already described in previous publications). Discussion on the other hand would significantly benefit from more application oriented, with an emphasis on the studied effect.

Designating the decimal separator with a point rather than a comma is generally preferred.

Citing articles from an open source rather than the original publication (e.g. reference [18]) is questionable.

Author Response

(The authors gave the same response as above.)

Round 2

Author Response

See the file

Reviewer 3 Report

The introduction and discussion sections still need significant improvement.

Author Response

See the file

Round 3

Reviewer 3 Report

The results section could use more extensive discussion.

Author Response

Reviuwer
